# A proportion of CD4+ T cells from patients with chronic Chagas disease undergo a dysfunctional process, which is partially reversed by benznidazole treatment

Elena Pérez-Antón[1], Adriana Egui[1], M. Carmen Thomas[1], Bartolomé Carrilero[2], Marina Simón[2], Miguel Ángel López-Ruz[3], Manuel Segovia[2], Manuel Carlos López[1]*

**1** Instituto de Parasitología y Biomedicina López-Neyra, Consejo Superior de Investigaciones Científicas; Granada, Spain, **2** Unidad Regional de Medicina Tropical, Hospital Virgen de la Arrixaca; El Palmar, Murcia, Spain, **3** Servicio de Medicina Interna, Hospital Virgen de las Nieves; Granada, Spain

* mclopez@ipb.csic.es

## Abstract

### Background

Signs of senescence and the late stages of differentiation associated with the more severe forms of Chagas disease have been described in the *Trypanosoma cruzi* antigen-specific CD4+ T-cell population. However, the mechanisms involved in these functions are not fully known. To date, little is known about the possible impact of benznidazole treatment on the *T. cruzi*-specific functional response of CD4+ T cells.

### Methodology/Principal findings

The functional capacity of CD4+ T cells was analyzed by cytometric assays in chronic Chagas disease patients, with indeterminate form (IND) and cardiac alterations (CCC) (25 and 15, respectively) before and after benznidazole treatment. An increase in the multifunctional capacity (expression of IFN-γ, IL-2, TNF-α, perforin and/or granzyme B) of the antigen-specific CD4+ T cells was observed in indeterminate *versus* cardiac patients, which was associated with the reduced coexpression of inhibitory receptors (2B4, CD160, CTLA-4, PD-1 and/or TIM-3). The functional profile of these cells shows statistically significant differences between IND and CCC (p<0.001), with a higher proportion of CD4+ T cells coexpressing 2 and 3 molecules in IND (54.4% *versus* 23.1% and 4.1% *versus* 2.4%, respectively). A significant decrease in the frequencies of CD4+ T cells that coexpress 2, 3 and 4 inhibitory receptors was observed in IND after 24–48 months of treatment (p<0.05, p<0.01 and p<0.05, respectively), which was associated with an increase in antigen-specific multifunctional activity. The IND group showed, at 9–12 months after treatment, an increase in the CD4+ T cell subset coproducing three molecules, which were mainly granzyme B+, perforin+ and IFN-γ+ (1.4% *versus* 4.5%).

### Conclusions/Significance

A CD4+ T cell dysfunctional process was detected in chronic Chagas disease patients, being more exacerbated in those patients with cardiac symptoms. After short-term

**Data Availability Statement:** All relevant data are within the manuscript and its Supporting Information files.

**Funding:** The authors have received funding from the following sources: MCL, grant SAF2016-81003-R from the Programa Estatal I+D+i (Agencia Estatal de Investigación-MINECO) and grant RD16/(0027/0005 from the Network of Tropical Diseases Research RICET (Instituto de Salud Carlos III)) and FEDER; MCT, grant SAF2016-80998-R from the Programa Estatal I+D+i (Agencia Estatal de Investigación - MINECO); MS, grant RD16/0027/0016 from the Network of Tropical Diseases Research RICET (Instituto de Salud Carlos III) and FEDER. The funders had no role in study design, data collection and analysis, decision to publish, or preparation of the manuscript.

**Competing interests:** The authors have declared that no competing interests exist.

benznidazole treatment (9–12 months), indeterminate patients showed a significant increase in the frequency of multifunctional antigen-specific CD4+ T cells.

## Author summary

*Trypanosoma cruzi* infection triggers several immune mechanisms in the host that do not result in a total clearance of the parasite, the persistence of which leads to the chronicity of Chagas disease. The mechanisms by which some chronic patients remain asymptomatic or become symptomatic are not entirely clear. The aim of the present manuscript is to study the CD4+ T cell population and its functional capacity in patients with different forms of chronic disease. The obtained results indicate that cells from indeterminate patients have an enhanced multifunctional profile, which is associated with the reduced expression of inhibitory molecules. CD4+ T cells from chronic patients with cardiac alterations show lower functional activity against specific antigens of the parasite and increased coexpression of inhibitory molecules. After benznidazole treatment, antigen-specific CD4+ T cells, especially those from indeterminate patients, are more likely to show a multifunctional profile and a decline in the coexpression of inhibitory receptors. These results allow us to make progress in clarifying the mechanisms that may influence disease progression and to realize the importance of antiparasitic treatment for the enhancement of the activity of the immune system.

## Introduction

Chagas disease is a globally neglected tropical disease that causes high social and economic burden in Latin America, where it is endemic [1]. The protozoan parasite *Trypanosoma cruzi* is the causal agent of Chagas disease. Currently, it is estimated that 7 million people are infected in countries in which Chagas disease is endemic and that approximately 10,000 deaths from this pathology occur each year [2]. In addition, as a result of migratory flows, Chagas disease has spread to non-endemic countries [3]. Parasite infection leads to several immune mechanisms in the host that do not result in a total clearance of the parasite, and its persistence causes the chronicity of Chagas disease [4]. Initially, the chronic phase of Chagas disease is apparently asymptomatic in the clinical stage that is called indeterminate. However, approximately 30–40% of asymptomatic patients develop fatal disorders after decades of *T. cruzi* infection, with cardiac disorders being the most frequent and most likely to be related to mortality [5]. Although the immune system plays a key role in the control of parasite infection and also perhaps in the maintenance of the absence of clinical symptoms, the mechanisms involved in these functions are not fully understood. An environment in which immune regulatory signals lead to the positive balancing of the immune system towards a Th1 response (IFN-γ and/or IL-2) *versus* a Th2 response (IL-4 and/or IL-10) [6,7]. In acute and chronic *T. cruzi* infections, the CD4+ T cells of Th1 phenotype appear to be the main responsible of a protective immunity by the expression of IFN-γ [8]. Moreover, the secretion of cytotoxic molecules, granzyme and perforin, have also shown to be crucial for the elimination of *T. cruzi* and for improving the outcome of the pathology [9]. It has been reported that antigen-specific T cells are actively involved in and mediate these immune mechanisms [10]. Furthermore, depletion of CD4+ and CD8+ T cells in murine models produces an increase in the parasite burden and the exacerbation of chronic pathology, which indicates the importance of these subsets of cells in the control of infection [11,12]. As a result, the study of Chagas disease in immunosuppressed

patients reveals the association of a defective immune system with the lack of control of *T. cruzi* infection, as is the case in patients with low CD4+ T-cell counts who have a high parasitic load, which is linked to coinfection with HIV [13]. Furthermore, in chronic viral infections, the exhaustion of the T cell functional response against the pathogen has been revealed to be due to persistent exposure to the pathogen antigens. The dysfunction of the antigen-specific T cell response is a gradual result of the loss of immune functions, such as lymphoproliferation, IL-2, TNF-α and IFN-γ cytokine production, cytotoxic activity, and the concomitant upregulation of the expression and coexpression of inhibitory receptors such as PD-1, TIM-3, LAG-3, CD160, 2B4, among others [14]. Recently, it has been reported that *T. cruzi*-specific CD8+ T cells undergo a similar dysfunctional cellular process during chronic infection. The CD8+ T cells of patients with chronic Chagas disease upregulate the expression of inhibitory receptors and, at the same time, lose the ability to produce cytokines and cytotoxic molecules with activity against *T. cruzi* antigens. Such dysfunction of the CD8+ T cell population was detected more readily in chronic patients with cardiac symptomatology than in those in the indeterminate stage of Chagas disease [15,16]. In this sense, several works have reported that in chronic Chagas disease the parasite-specific T-cell responses are modulated by the antiparasitic treatment. Thus, changes in the expression level, type of secreted cytokines and phenotype of the antigen-specific T cells have been described after treatment [17–20]. Thus, modifications in *T. cruzi*-specific T-cell responses could be used as early indicators of response to treatment, even at cases where a cure is not achieved [21]. Furthermore, it was reported that antiparasitic treatment partially reversed this process by increasing the multifunctional capacity of antigen-specific CD8+ T cells and decreasing the coexpression of inhibitory receptors in these cells [16,22]. Signs of senescence, the later stages of differentiation, the effects of immunoregulatory mechanisms and the expression of inhibitory receptors associated with more severe forms of Chagas disease have also been described in the CD4+ T-cell population [4,23,24]. However, compared to that in CD8+ T cells, little is known about the dysfunction process in CD4+ T cells in chronic *T. cruzi* infection. Thus, the aim of this work was to evaluate CD4+ T cells from patients with chronic Chagas disease in both the indeterminate form of the disease and in patients with cardiac alterations and to determine the expression and coexpression of inhibitory molecules and how the antigen-specific multifunctional capacity of these cells is affected. Furthermore, we evaluated the possible impact of benznidazole chemotherapy on the *T. cruzi*-specific functional response of CD4+ T cells and the expression of coinhibitory molecules by these cells.

## Study population, material and methods

### Ethics statement

The Ethics Committees of the Consejo Superior de Investigaciones Científicas (CSIC), Spain (Reference: 094/2016), and Virgen de la Arrixaca Hospital, Murcia-Spain (Reference: 25/05/ 2016), approved the protocols used in this study. Signed informed consent was obtained from all individuals before their inclusion in the study.

### Description of the study cohort

The study was carried out by evaluating 40 patients with Chagas disease who were diagnosed according to the WHO criteria based on two conventional serological tests (Chagas ELISA, Ortho Clinical Diagnosis, and Inmunofluor Chagas, Biocientífica, Argentina) at the Virgen de la Arrixaca Hospital, Murcia, Spain and at the Virgen de las Nieves University Hospital, Granada, Spain. All included patients were in the chronic phase of Chagas disease (cChD). Patients were considered to be at the indeterminate phase (IND; n = 25) when they were seropositive with no evidence of cardiac disorder (G0 following Kuschnir classification based on

**Table 1. PCR for parasite detection in patients with Chagas disease pre and post benznidazole treatment.**

| Patients group | Age (years) | | Sex | Percentage of patients with PCR-positive | | |
|---|---|---|---|---|---|---|
| | Mean ± SD | Range | (females / males) | Before BNZ treatment | Months after treatment | |
| | | | | | 9–12 (T1) | 24–48 (T2) |
| Indeterminate | 35 ± 8.7 | 22–52 | 15 / 10 | 54.6% | 5.3% | 0% |
| (n = 25) | | | | (12 out of 22 patients) | (1 out of 19 patients) | (0 out of 20 patients) |
| Cardiac | 39.8 ± 10.3 | 25–54 | 7 / 8 | 57.1% | 0% | 0% |
| (n = 15) | | | | (8 out of 14 patients) | (0 out of 13 patients) | (0 out of 13 patients) |

clinical criteria and radiological, electrocardiographic, and transthoracic echocardiography analyses). Patients who manifested cardiac pathology (CCC; n = 15) were also classified based on the above referred Kuschnir classification (9 patients were in stage G1, 5 in stage G2 and 1 in stage G3), according to clinical criteria and radiological, electrocardiographic, and transthoracic echocardiography analyses [25]. Patients who showed evidence of gastrointestinal damage, presented any co-morbidity, such as diabetes, hypertension, etc., were excluded from the study. In addition, 12 healthy donors (HD) were evaluated in this study. All included patients were residents of a non-endemic area in Spain and had never been treated with anti-parasitic drugs before joining the study. Ninety-five percent (38 out of 40) of the patients came from Bolivia and the other two from Paraguay and Argentina. The data referring to the age range and sex distribution are detailed in Table 1. Untreated patients were not included as the Ethics Guideline from the Ethics Committee of the Virgen de la Arrixaca Hospital makes obligatory to offer and to provide the treatment to all patients with positive serology for Chagas disease.

## Peripheral blood samples

The extraction of peripheral blood samples from each subject included in the study was performed by intravenous puncture. A volume of 30 mL of blood was collected for each sample. Blood samples from each patient were collected at 3 different times: before treatment (T0), after 9–12 months (T1), and after 24–48 months (T2) of treatment with benznidazole (BNZ). The patients were treated with chemotherapy, which consisted of the administration of 5 mg/kg/d benznidazole for 60 days [5], and were clinically monitored during the entire study period at the corresponding hospital.

## Peripheral blood mononuclear cell isolation

Peripheral blood mononuclear cells (PBMCs) from the samples obtained from each subject included in the study at each selected time point were always obtained following the same protocol. The isolation was carried out by using density gradient centrifugation as previously described [26]. The purified PBMCs were diluted in inactivated fetal bovine serum (iFBS) (Gibco, Grand Island, NY) with 10% DMSO. Liquid nitrogen was used for the cryopreservation of the samples prior to use. All blood samples were collected and maintained to room temperature. PBMCs were purified between 16 and 18 hours after collection and subsequently frozen in liquid nitrogen following the same protocol. The pretreatment and posttreatment samples from each one of the patients were thawed and processed and analyzed at the same time, verifying that viability was greater than 80% in all cases.

## DNA extraction and PCR for parasite detection

Genomic DNA was purified from peripheral blood from patients and PCR amplification for parasite detection was performed at the hospital of origin of each patient, as previously

described by Murcia et al. [27]. Briefly, genomic DNA was extracted using the Maxwell 16 Blood DNA Purification Kit (Promega Biotech Iberica). PCR detection of the 330 bp variable regions of the *T. cruzi* kinetoplast minicircle genome was carried out as described by Murcia et al [27]. Negative controls were included in each PCR. Amplified products were resolved in 2% agarose gels and visualized after ethidium bromide staining by UV exposure.

## Isolation of *Trypanosoma cruzi* soluble antigens

*T. cruzi* soluble antigens (*Tc*SA) were required to carry out the *in vitro* stimulation performed in this study. For this reason, rhesus monkey kidney epithelial cells (LLC-MK2 line; CCL-7, Manassas, VA) were cultured in RPMI-1640 supplemented with 2 mM L-glutamine (Gibco), 10% iFBS and 50 μg/mL gentamicin (Thermo Fisher Scientific, Waltham, Massachusetts, USA) at 37˚C in 5% $CO_2$ according to a previously described protocol [28]. Infection of the LLC-MK2 semiconfluent monolayers was performed with the trypomastigote form of the Y strain of *T. cruzi* (MHOM/BR/1950/Y; DTU II). Trypomastigotes forms were recovered from the infected-culture supernatants after 96 h post-infection and subsequent days the amastigotes were collected from the infected-culture supernatants. Potential cellular detritus were eliminated by centrifugation at 1,000 rpm for 4 min and the supernatants centrifuged at 2,500 rpm for collection of amastigote and trypomastigote forms, respectively. Soluble total proteins were extracted by resuspension of parasites (50% of each form) in lysis buffer (50 mM Tris-HCl at pH 7.4, 0.05% Nonidet P-40, 50 mM NaCl, 1 mM phenylmethylsulfonyl fluoride (PMSF), 1 μg/mL leupeptin) and sonication 3 times with pulses of 50–62 KHz for 40-s time intervals of 20 s. Protein concentration was determined using the micro BCA protein assay kit (Thermo Fisher Scientific) and visualized by SDS-PAGE after Coomassie blue staining (Gibco).

## Labeling with specific antibodies for multiparametric flow cytometry assays

For staining of the cell surface, the following conjugated antibodies were used: CD3-Pacific Blue (clone UCHT1), CD4-Alexa Fluor 700 (clone RPA-T4), CD8-APC-H7 (clone SK1), CD160-Alexa Fluor 647 (clone BY55), TIM-3-PE-CF594 (clone 7D3), 2B4-FITC (clone 2–69) (BD Biosciences, San Jose, CA), and PD-1-PE (clone J105) (Thermo Fisher Scientific). Intracellular staining was performed with the following antibodies: CTLA-4-PE-Cy5 (clone BNI3), granzyme B-PE-CF594 (clone GB11), IFN-γ-PE-Cy7 (clone B27), IL-2-APC (clone MQ1-17H12), TNF-α-Alexa Fluor 488 (clone MAb11) (BD Biosciences, San Diego, CA), and perforin-PE (clone B-D48) (Abcam, Cambridge, UK). All conjugated antibodies were titrated as previously reported [29]. The multicolor panels used for the flow cytometry assays were utilized according to the fluorescence minus one control (FMO) method as recommended [29]. In addition, the following purified NA/LE antibodies were used for cell culture: CD28 (clone CD28.2) and CD49d (clone 9F10) (BD Biosciences).

## Flow cytometry assays for functional characterization after stimulation with *Trypanosoma cruzi* soluble antigens

The stimulation of PBMCs with *Tc*SA (1 μg/mL) was carried out for 10 h at 37˚C in a humidified atmosphere with 5% $CO_2$ as reported in previous studies [16,29–32]. Due to the limited number of cells isolated from the blood sample collected from each patient (following the protocol approved by the Ethics Committee) together to the number of cells required to analyze all the biomarkers under study, the stimulation of PBMCs was performed only at one time (10

h). Cells were cultured in RPMI-1640 with 2 mM L-glutamine, 10% iFBS and 50 μg/mL genta-micin. The cell concentration was 1x10^6 cells/mL, and cells were incubated in the presence of anti-CD28 (1 μg/mL) and anti-CD49d (1 μg/mL) as previously reported [29,30,32] and were either stimulated with *Tc*SA (1 μg/mL) or without stimulation (regarded as the basal response).

**Intracellular staining to evaluate the production of cytokines and cytotoxic molecules.** In addition to the considerations described above for the flow cytometry assays, to detect the production of cytokines and cytotoxic molecules, and after stimulation with *Tc*SA (1 μg/mL) for 1 h at 37˚C [30], the cells were cultured for additional 9 h in the presence of bre-feldin A (1 μg/mL) and monensin (2 μM) (BD Bioscience) [31]. The additional steps that were followed for the detection of the IFN-γ, IL-2, TNF-α, granzyme B and perforin molecules produced by CD4+CD8- T cells were detailed in a previously described protocol [31]. Briefly, PBMCs were stained using LIVE/DEAD Fixable Aqua Dead Cell Stain kit (Invitrogen, Eugene, OR). Following surface staining using antibodies against the CD3, CD4 and CD8 molecules, the cells were washed, fixed and permeabilized using Cytofix/Cytoperm (BD Biosciences). Intracellular staining was performed using conjugated antibodies against granzyme B, IFN-γ, IL-2, TNF-α, and perforin at the same time. Details of the each specific antibody including clone, conjugated fluorochromes and references are referred above. The gating strategy used in the flow cytometry analysis was included in the S1 Fig. CD4+ T cell selection was performed after selection of CD3+ cells and the exclusion of both CD8+ and CD4+CD8+ double-positive T cells. Exclusion was performed in order to not include CD4+CD8+ T cells, which are described as an independent population with a high multifunctional capacity and high expression of activation markers compared to single-positive T cell subsets [31,33]. The samples were acquired employing the FASCAria III cytometer (BD Biosciences). At least 100,000 lympho-cytes, according to their FCS/SSC parameters, were acquired for each sample and each condi-tion, using the FASCDiva software (BD Bioscence). The data files were subsequently analyzed to the percentages of expression of each molecule. The percentage of expression of each mole-cule was calculated using FlowJo 9.3.2 software (Tree Star, Ashland, OR) by subtracting the percentage of the background (condition without stimulus plus CD28 and CD49d) from the data obtained after the condition of stimulation with *Tc*SA plus CD28 and CD49d. In addition, coexpression studies were carried out using a Boolean gates strategy. Positivity for each marker was selected based on the fluorescence minus one (FMO) control, and the unstained control in each antibody panel evaluated in the multiparametric studies carried out.

**Surface staining to evaluate the expression of inhibitory receptors.** Staining was per-formed to detect the expression and coexpression of different inhibitory receptors in the popu-lation of CD4+CD8- T cells. The assessed inhibitory receptors were 2B4, CD160, PD-1, and TIM-3. The protocol that was carried out was similar to that previously described [31]. Briefly, PBMCs were stained using LIVE/DEAD Fixable Aqua Dead Cell Stain kit (Invitrogen, Eugene, OR). Subsequently, surface staining was performed by incubating the cells with anti-CD3, anti-CD4, anti-CD8, anti-2B4, anti-CD160, anti-PD-1 and anti-TIM-3 antibodies. Cells were washed, fixed and permeabilized for subsequent intracellular staining with anti-CTLA-4 anti-body. Cells were acquired in a similar way to the previous multi-color panel in this study. The data files were analyzed using FlowJo 9.3.2 software to calculate the expression percentages of each molecule and the mean fluorescence intensity (MFI) in each case.

## Statistical analysis

The statistical analysis used for the comparisons between groups and between the different time points that were tested as well as the generation of the respective graphic representations

were performed using GraphPad Prism version 6.0 software (GraphPad Software, San Diego, CA). The Shapiro-Wilk normality test was performed to evaluate the type of data distribution. Comparisons between groups (patients *versus* healthy donors or indeterminate patients [IND] *versus* cardiac patients [CCC]) were made using the nonparametric Mann-Whitney U test or the parametric unpaired t test with Welch's correction according to the data distribution results. The nonparametric Wilcoxon test and the parametric paired t test were used to compare the different study time points. The Spearman's rank correlation coefficients ($\rho$) and nominal p-values were calculated to determine significant correlations between the evaluated parameters. In addition, SPICE software version 5.3 (National Institutes of Health, Bethesda, MD) was used to compare pie charts by generating 10,000 permutations. The values of p denoted as statistically significant were those values of p lower than 0.05 ($p < 0.05$). The symbols used for the different p values are $p < 0.05$ (*), $p < 0.01$ (**), $p < 0.001$ (***) and $p < 0.0001$ (****). The whiskers of the box-plot, which was generated with GraphPad software, represent all the values (from minimum to maximum), and box represents the 25th to the 75th percentile. In addition, evaluation of the variability of the data was performed by analyzing the outliers of the variables under study. The obtained results indicated that the variability observed among the values was not associated to a particular patient.

## Results

### Expression of inhibitory molecules in the population of circulating CD4+ T cells in patients with chronic Chagas disease

The expression of the inhibitory receptors 2B4, CD160, CTLA-4, PD-1 and TIM-3 in the CD4+ T cell population was determined by following the steps described in the "Material and methods" section of this manuscript. Samples from 38 patients with chronic Chagas disease (cChD) [25 indeterminate patients (IND) and 13 patients with cardiac symptomatology (CCC)] and 12 healthy donors (HD) were evaluated. Thus, the frequency of CD4+ T cells expressing each of the evaluated inhibitory receptors and the given mean fluorescence intensity (MFI) for each marker were calculated for cChD (IND and CCC) and HD (Fig 1). A higher frequency of CD4+ T cells expressing 2B4 ($p < 0.0001$), CTLA-4 ($p < 0.05$), PD-1 ($p < 0.0001$) and TIM-3 ($p < 0.01$) in cChD was found compared to that in HD (Fig 1A). Likewise, according to the MFI, higher levels of expression of the molecules 2B4 ($p < 0.05$), CTLA-4 ($p < 0.01$) and PD-1 ($p < 0.0001$) were also detected in cChD compared with those in HD (Fig 1B). When the patients were evaluated according to the degree of pathology (IND and CCC), the greatest differences in the frequencies of CD4+ T cells expressing each inhibitory receptor under study were observed between CCC and HD, and the significance of the differences were as follows: 2B4 ($p < 0.0001$), CD160 ($p < 0.01$), CTLA-4 ($p < 0.0001$), PD-1 ($p < 0.0001$) and TIM-3 ($p < 0.01$) (Fig 1A). In addition, the frequency of CD4+CD160+ ($p < 0.05$) and CD4+CTLA-4+ T cells ($p < 0.0001$) was higher in CCC than in IND. Furthermore, the expression levels of each inhibitory receptor calculated according to the MFI (Fig 1B) were higher for the 2B4 ($p < 0.01$), CTLA-4 ($p < 0.001$) and PD-1 ($p < 0.0001$) inhibitory receptors in CD4+ T cells from CCC compared to those in CD4+ T cells from HD. The results also showed statistically significant differences ($p < 0.05$) in the expression of 2B4, CTLA-4 and TIM-3 in IND *versus* those in CCC. Additionally, IND showed a significant difference due to the increased expression of PD-1 with respect to HD ($p < 0.001$) (Fig 1B).

The coexpression of the inhibitory receptors under study (2B4, CD160, CTLA-4, PD-1 and TIM-3) was analyzed. The results shown in Fig 2 indicate that the level of the coexpression of two, three, four and five inhibitory receptors by the CD4+ T cell population was higher in cChD than in HD ($p < 0.0001$, $p < 0.0001$, $p < 0.001$ and $p < 0.01$, respectively). Furthermore,

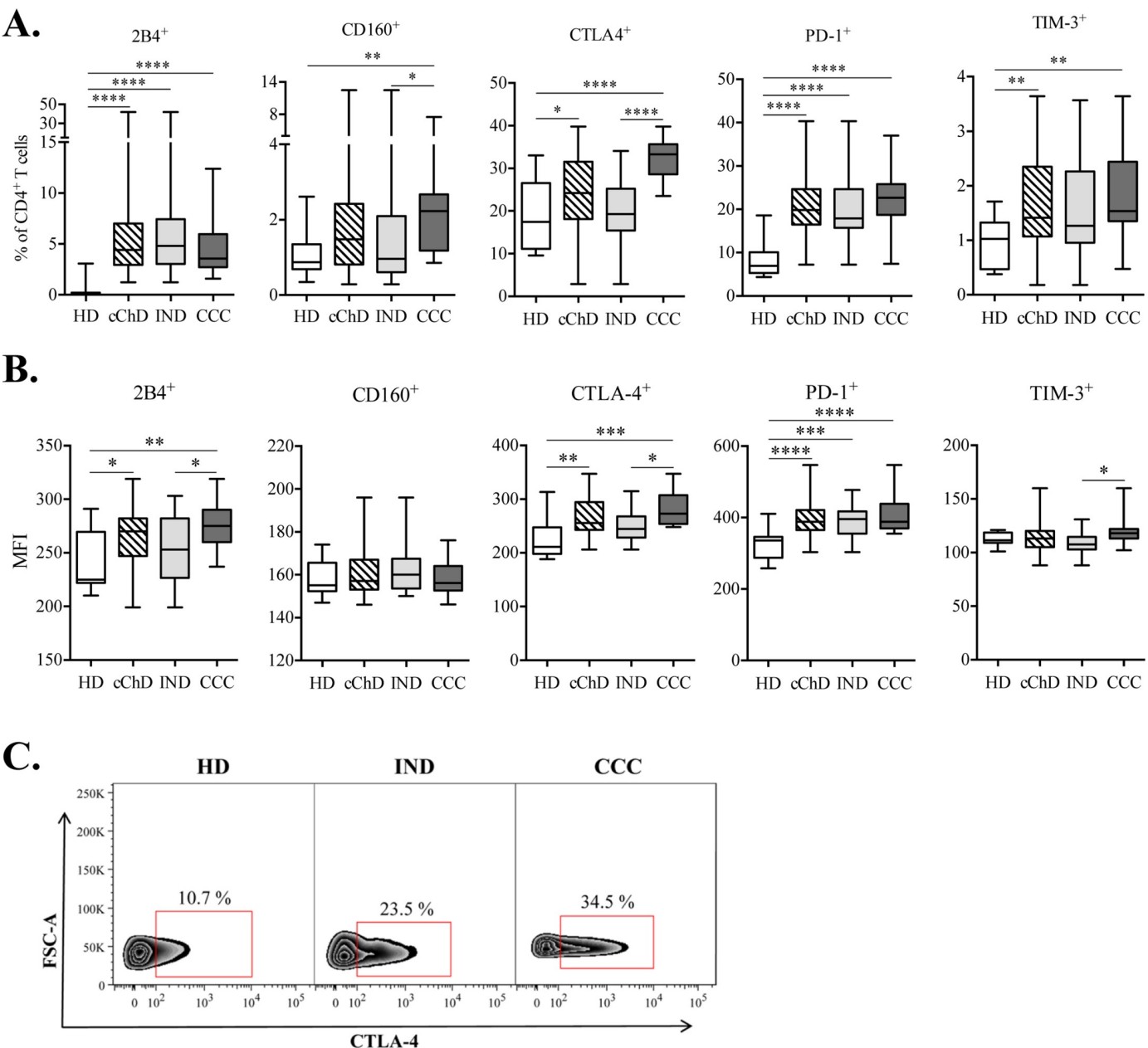

**Fig 1. Expression of inhibitory receptors in the CD4⁺ T cell population in chronic Chagas disease patients, including indeterminate patients and patients with cardiac alterations. (A)** The frequencies of CD4⁺ T cells expressing 2B4, CD160, CTLA-4, PD-1 or TIM-3 in 38 chronic Chagas disease patients (cChD), including 25 indeterminate patients (IND) and 13 patients with cardiac manifestations (CCC), were calculated for each sample and compared to those frequencies in 12 healthy donors (HD). **(B)** The expression levels of the inhibitory receptors under study in CD4⁺ T cells were measured according to the value of the mean fluorescence intensity (MFI) and represented in a box-plot for each marker in each study group (HD, cChD, IND and CCC). The analysis was performed by flow cytometry and data calculated using the FlowJo 9.3.2 software. Cell cultures were *in vitro* stimulated with *T. cruzi* soluble antigens (*Tc*SA) for 10 h in the presence of CD28 and CD49d co-stimulators. The range of values represented by the box and the whiskers includes the values from the minimum to the maximum. Statistical analyzes were performed using the Mann-Whitney U test. The p values are represented by asterisks as follows: p<0.05 (*), p<0.01 (**), p<0.001 (***) and p<0.0001 (****). **(C)** Example of representative plot of the frequency of CD4⁺ T cells that express CTLA-4 in a healthy donor (HD), an indeterminate (IND) and a patient with cardiac symptomatology (CCC).

higher coexpression of inhibitory receptors was found in CD4⁺ T cells from CCC than in those from IND, with a statistically significant difference in the coexpression of two and five inhibitor molecules (p<0.05) (Fig 2).

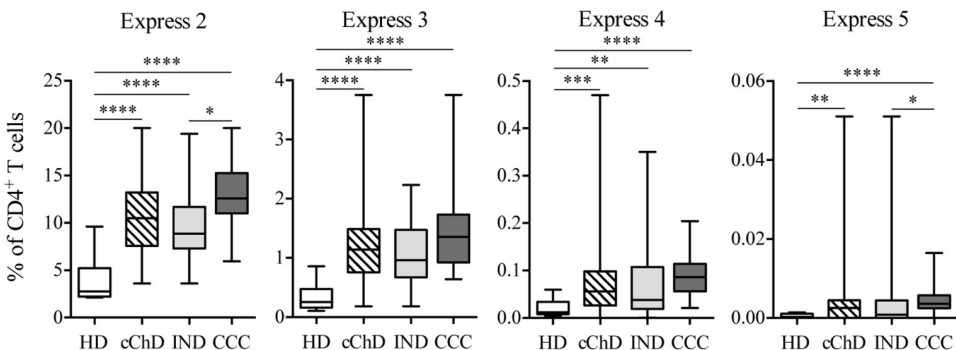

**Fig 2. Frequency of the coexpression of inhibitory receptors in CD4+ T cells from chronic Chagas disease patients.**
Analysis of the CD4+ T cells coexpressing any of the inhibitory receptors under study (2B4, CD160, CTLA-4, PD-1 and TIM-3) in chronic Chagas disease patients (cChD) [indeterminate (IND) and with cardiac symptomatology (CCC)] in comparison with that in healthy donors (HD). The size of the study population was 38 for cChD (25 IND and 13 CCC) and 12 for HD. The coexpression analyses of the five inhibitory receptors under study were carried out using a Boolean gates strategy using FlowJo 9.3.2 software. The box and whiskers of the graph represent all the values (from minimum to maximum). Statistical analyzes were performed using the Mann-Whitney U test. The p values shown by the asterisks are as follows: p<0.05 (*), p<0.01 (**), p<0.001 (***) and p<0.0001 (****).

## Impact of benznidazole on the expression and coexpression of inhibitory receptors in the population of CD4+ T cells from chronic Chagas disease patients

The impact of BZN treatment on the expression and coexpression of inhibitory receptors in the CD4+ T cell population from 40 cChD (25 IND and 15 CCC) was evaluated. Thus, the percentage of CD4+ T cells that express 2B4, CD160, CTLA-4, PD-1 and TIM-3 was measured in patients before treatment (T0) and at 9–12 months (T1) and 24–48 months (T2) after BNZ administration (Fig 3). The obtained results showed that the frequencies of CD4+ T cells expressing 2B4, CD160 and PD-1 decreased after 9–12 and 24–48 months of treatment in cChD, with statistical significance (2B4, p<0.05 and p<0.001, respectively; CD160, p<0.05 and p<0.01, respectively; PD-1, p<0.05 and p<0.0001, respectively). In contrast, the frequencies of CD4+CTLA-4+ and CD4+TIM-3+ T cells increased after BNZ treatment. Thus, the percentages of both subsets (CD4+CTLA-4+ and CD4+TIM-3+ T cells) were statistically decreased before treatment administration compared to those after 9–12 and 24–48 months of treatment (p<0.05 and p<0.01, respectively) (Fig 3). The frequencies of CD4+ T cells expressing the 2B4, CD160, CTLA-4, PD-1 or TIM-3 inhibitory receptors were also analyzed in IND and CCC before and after treatment (gray box plots, Fig 3). The obtained results show that the frequencies of CD4+ T cells expressing 2B4, CD160 and PD-1 decreased significantly after 24–48 months of BNZ treatment in IND (p<0.01, p<0.05, p<0.001; respectively). Conversely, a significant increase in the frequency of CD4+CTLA-4+ and CD4+TIM-3+ T cells after 9–12 months of treatment (p<0.01 and p<0.05, respectively) and after 24–48 months (p<0.01) was observed in IND (light gray box plots, Fig 3). After treatment (at 9–12 and 24–48 months), CD4+ T cells in CCC showed a decrease in the expression of PD-1 (p<0.05) (dark gray box plot, Fig 3). Furthermore, the results obtained after a Spearman correlation analysis including all the Chagas disease patients showed a positive correlation between the modification of the expression level of the five inhibitory receptors evaluated and the time-course after treatment (S1 Table).

Furthermore, the impact of BNZ treatment on the level of the coexpression of inhibitory receptors assessed in CD4+ T cells from cChD was analyzed in IND *versus* CCC (Fig 4). A significant decrease in the frequencies of CD4+ T cells that coexpressed 2, 3 and 4 inhibitory

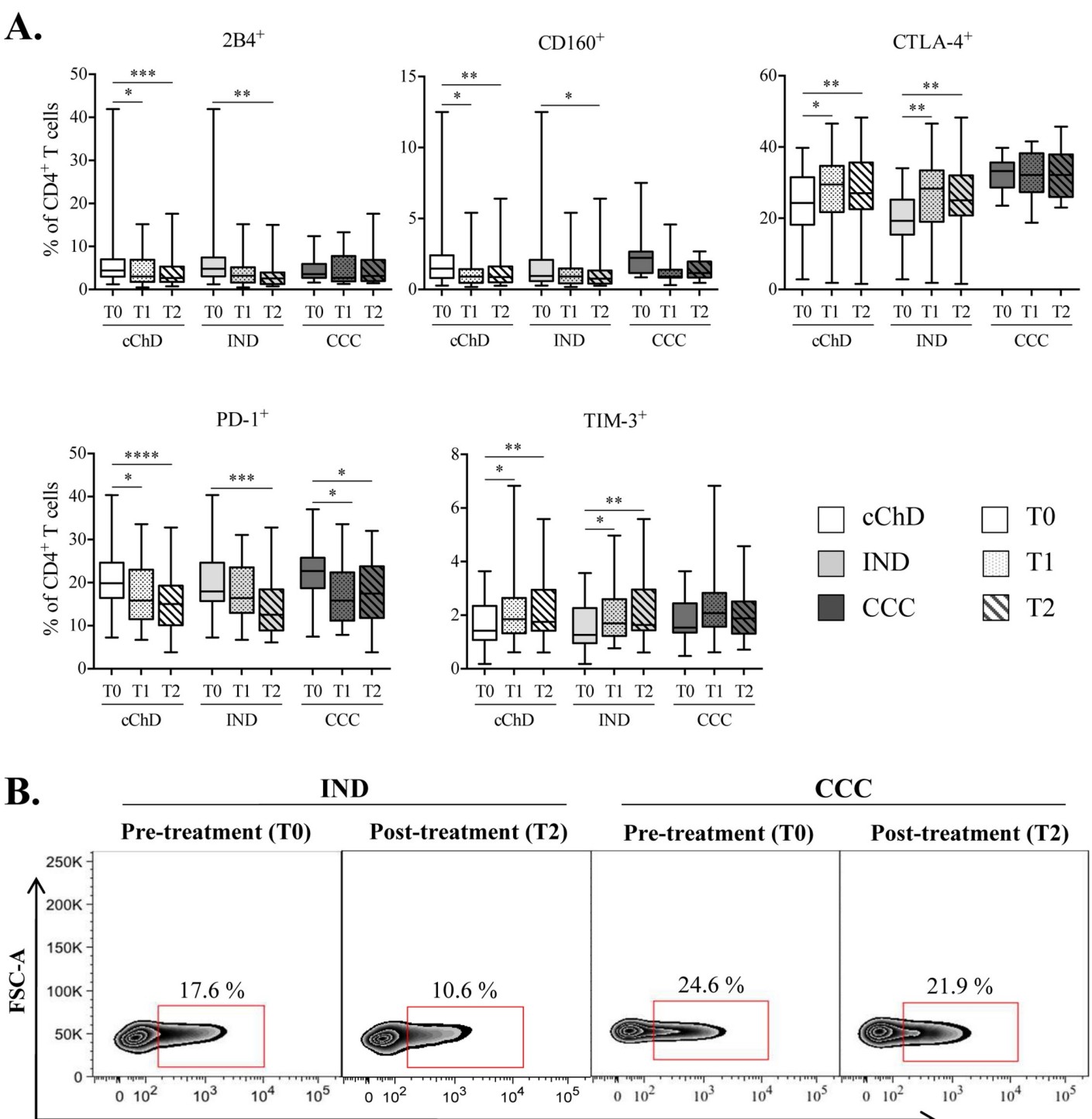

**Fig 3. Impact of benznidazole treatment on the expression of inhibitory receptors in CD4+ T cells from chronic patients with Chagas disease. (A)** The evaluated inhibitory receptors were 2B4, CD160, CTLA-4, PD-1 and TIM-3. The study was carried out in 40 chronic Chagas disease patients (cChD), 25 indeterminate patients (IND) and 15 cardiac patients (CCC). The time points of the study were pretreatment (T0) and two time points posttreatment (after 9–12 months (T1) and after 24–48 months (T2)). Evaluation of the frequencies of CD4+ T cells expressing each inhibitory receptor under study was carried out before and after treatment in all chronic ChD patients under study (cChD) and separately in each group of patients (indeterminate (IND) and cardiac (CCC)). The analysis was performed by flow cytometry and data calculated using the FlowJo 9.3.2 software. Cell cultures were *in vitro* stimulated with *T. cruzi* soluble antigens (*Tc*SA) for 10 h in the presence of CD28 and CD49d co-stimulators. The range of values represented by the box and the whiskers includes the values from the minimum to the maximum. Statistical analyzes were performed using the Wilcoxon test. The p values are represented by asterisks as follows: p<0.05 (*), p<0.01 (**), p<0.001 (***) and p<0.0001 (****).

**(B)** Representative plot showing the frequency of CD4⁺ T cells expressing PD-1 previously (T0) and after 24–48 months of the treatment (T2) in an indeterminate patient (IND) and a patient with cardiac symptomatology (CCC).

receptors was observed in IND after 24–48 months of treatment ($p<0.05$, $p<0.01$ and $p<0.05$, respectively). Meanwhile, in the CCC group, a statistically significant decrease was found in the frequency of CD4⁺ T cells that coexpressed 2 inhibitory receptors at 9–12 and 24–48 months after treatment (both values, $p<0.05$) (Fig 4). In CCC, the frequencies of CD4⁺ T cells expressing 3 and 4 receptors showed a downward trend that did not reach statistical significance (Fig 4).

## Modulation of the CD4⁺ T cell subsets expressing inhibitory receptors from indeterminate Chagas disease patients after treatment

The results shown in Fig 5 show that at 9–12 and 24–48 months after treatment, there were significant decreases in the CD4⁺ T cell subsets that coexpressed three inhibitory receptors: 2B4⁺CD160⁺PD-1⁺ ($p<0.01$ and $p<0.001$), 2B4⁺CTLA-4⁺PD-1⁺ ($p<0.01$ and $p<0.0001$), and CD160⁺PD-1⁺TIM-3⁺ ($p<0.05$ and $p<0.01$). Moreover, the CD4⁺ T cell subset coexpressing four inhibitor receptors (2B4⁺CD160⁺CTLA-4⁺PD-1⁺TIM-3⁻) presented a decrease at 24–48 months after treatment ($p<0.01$). Likewise, other CD4⁺ T cell subsets (2B4⁺CD160⁺CTLA-4⁺PD-1⁻TIM-3⁻ and 2B4⁺PD-1⁺CD160⁻CTLA-4⁻TIM-3⁻) were also decreased at 24–48 months after treatment ($p<0.05$ and $p<0.0001$, respectively) (Fig 5). Conversely, a statistically significant increase was observed after 9–12 ($p<0.01$) and 24–48 ($p<0.001$) months of treatment in the subset of cells that expressed only the CTLA-4 inhibitor receptor (CD4⁺CTLA-4⁺2B4⁻CD160⁻PD-1⁻TIM-3⁻) (Fig 5). Likewise, after treatment, an increase in the subset of cells that coexpressed CTLA-4 and TIM-3 (CD4⁺CTLA-4⁺TIM-3⁺2B4⁻CD160⁻PD-1⁻) was also detected at 9–12 months ($p<0.001$) and at 24–48 months ($p<0.0001$) (Fig 5). Moreover, increases in the subsets of CD4⁺ T cells expressing CTLA-4⁺PD-1⁺TIM-3⁺2B4⁻CD160⁻ ($p<0.05$); CD160⁺CTLA-4⁺TIM-3⁺2B4⁻PD-1⁻ ($p<0.01$); 2B4⁺CTLA-4⁺TIM-3⁺CD160⁻PD-1⁻ ($p<0.05$) and CD160⁺CTLA-4⁺PD-1⁺TIM-3⁺2B4⁻ ($p<0.05$) were observed after 9–12 and/or 24–48 months of treatment (Fig 5). Therefore, the percentages of

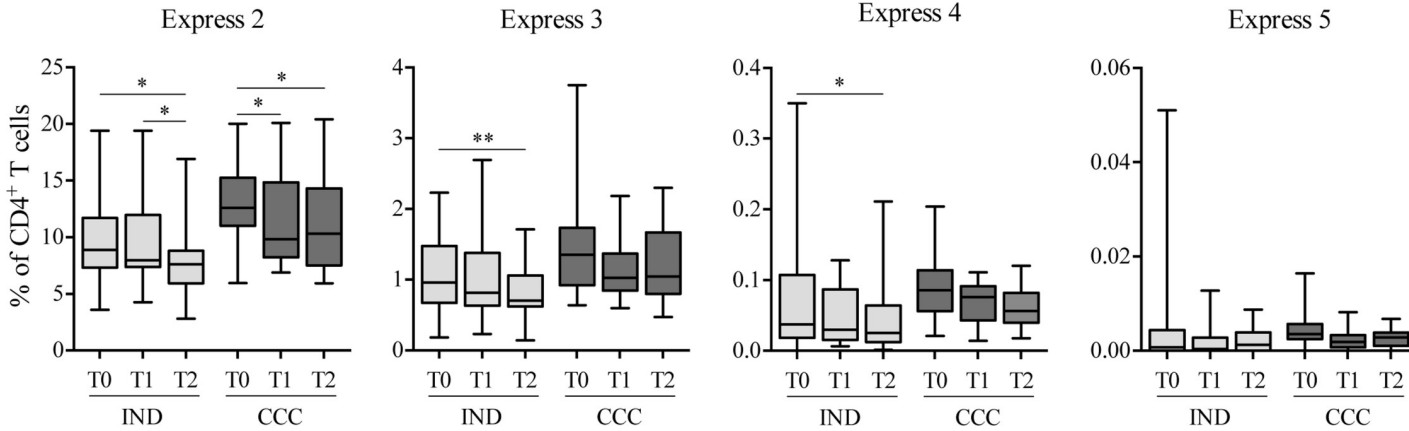

**Fig 4. Impact of benznidazole treatment on the level of the coexpression of inhibitory receptors in CD4⁺ T cells, as evaluated in chronic Chagas disease patients.** The inhibitory receptors evaluated in the coexpression study were 2B4, CD160, CTLA-4, PD-1 and TIM-3. The box-plot graphs show the percentages of CD4⁺ T cells coexpressing 2, 3, 4 or 5 inhibitory receptors, as evaluated before and after treatment with benznidazole, in indeterminate patients (IND, n = 25) and cardiac patients (CCC, n = 15). The tested time points were T0, pretreatment; T1, 9–12 months after treatment administration; and T2, after 24–48 months. The coexpression studies of the five inhibitory receptors under study were carried out using a Boolean gates strategy using FlowJo 9.3.2 software. The range of values represented by the box and the whiskers includes the values from the minimum to the maximum. Statistical analyzes were performed using the Wilcoxon test. The p values are represented by asterisks as follows: $p<0.05$ (*) and $p<0.01$ (**).

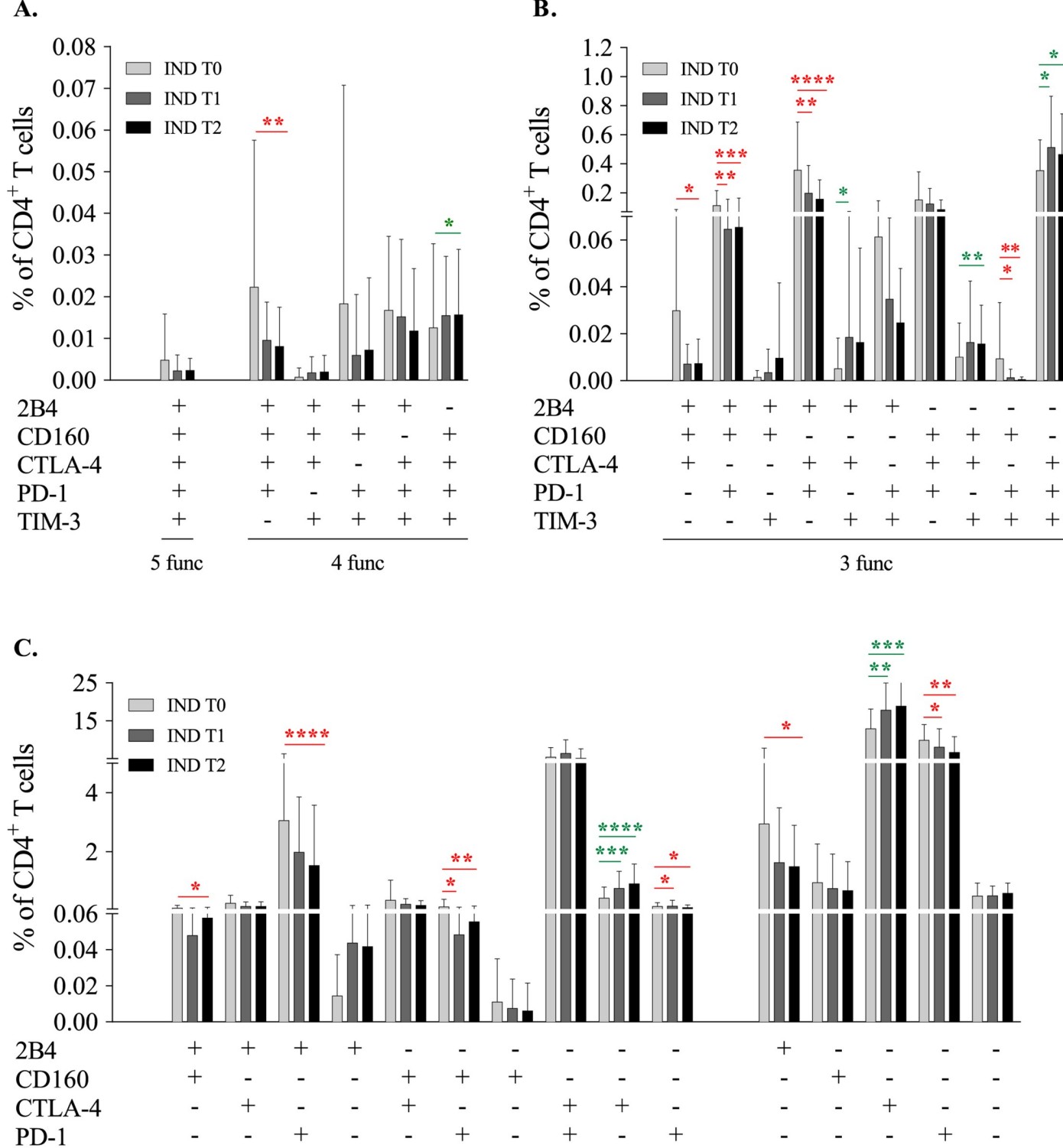

**Fig 5. Impact of treatment on subsets of CD4+ T cells with specific coexpression of different inhibitory receptors in indeterminate patients.** Percentages of CD4+ T cell subsets that coexpressed the different combinations of inhibitory receptors under study (2B4, CD160, CTLA-4, PD-1 and TIM-3) in 25 chronic indeterminate Chagas disease patients (IND). The values represent the frequencies before benznidazole treatment (T0, light gray columns), after 9–12 months of

treatment (T1, dark gray column) and after 24–48 months of treatment (T2, black columns). Frequency of CD4+ T cells coexpressing **(A)** five and four**, (B)** three**, (C)** two and one inhibitory receptors. The coexpression analyses of the five inhibitory receptors under study were carried out employing a Boolean gates strategy using FlowJo 9.3.2 software. The subsets of cells that after benznidazole treatment reduced or increased the percentages of expression with statistically significance are indicated with red and green asterisks, respectively. Statistical analyses were performed using the Wilcoxon test. The p values are represented by asterisks as follows: p<0.05 (\*), p<0.01 (\*\*), p<0.001 (\*\*\*) and p<0.001 (\*\*\*\*).

those cellular subsets that showed an increase posttreatment were small and mainly represented cells expressing a low number of inhibitory receptors (1 or 2), while those cells showing a reduction in their percentage after treatment were greater in number and coexpressed several inhibitory receptors (mainly 3).

## Multifunctional response of *Trypanosoma cruzi*-specific CD4+ T cells in patients with chronic Chagas disease

The antigen-specific functional and multifunctional response against *Tc*SA by CD4+ T cells was evaluated by measuring the production and coproduction of several cytokines and cytotoxic molecules (IFN-γ, IL-2, TNF-α, granzyme B and perforin) as described in the "Material and methods" section. This study was performed to evaluate the response of the CD4+ T cell population in 32 cChD (19 IND and 13 CCC), and the results were analyzed and represented using pie charts (Fig 6, panel T0). The functional profile was significantly different in the IND and CCC groups (p<0.001). A greater proportion of multifunctional antigen-specific CD4+ T cells was found in IND than in CCC, and a higher proportion of CD4+ T cells coexpressing 2 and 3 of the studied molecules was found in IND than in CCC (54.4% *versus* 23.1%; 4.1% *versus* 2.4%; respectively). In fact, a statistically significant increase in the percentage of antigen-specific CD4+ T cells that coproduced three molecules was detected in IND *versus* CCC (p<0.05) (S2 Fig). Monofunctional cells represented 74.0% of the *T. cruzi*-specific CD4+ T cells in CCC, whereas in IND, they did not exceed 41.1% of the total. It was also observed that the proportion of CD4+ T cells coexpressing both cytotoxic molecules under study (CD4+granzyme B+perforin+) was higher in IND (49.8%) than in CCC (3.1%). However, the CCC group of patients presented a higher proportion of CD4+ T cells expressing only the cytotoxic molecule granzyme B (CD4+granzyme B+perforin-) (45.5%) or perforin (CD4+granzyme B-perforin+) (14.4%) compared with the IND group (3.9% and 3.1%; respectively). Additionally, a greater proportion of CD4+ T cells that simultaneously expressed IFN-γ and granzyme B (CD4+IFN-γ+granzyme B+) was found in CCC (15.8%), whereas in IND, the proportion was no greater than 0.6% (Fig 6, panel T0).

To assess the impact of the treatment on the functional response of antigen-specific CD4+ T cells from IND and CCC chronic Chagas disease patients, the functional and multifunctional capacity (production or coproduction of IFN-γ, IL-2, TNF-α, granzyme B and perforin) of these cells was analyzed before and after treatment. As observed in the median values shown in S3 Fig, the percentage of CD4+ T cells from IND patients showed an increase in the expression of IFN-γ shortly after treatment (9–12 months) which decreased at 24–48 months after treatment. Increased secretion of granzyme B was also detected in IND after treatment, particularly during a prolonged period after treatment (24–48 months). In patients with CCC, a decrease in the percentage of cells expressing IL-2 was observed at 24–48 months after treatment (p<0.05). In addition, a decrease in the percentage of cells expressing IFN-γ and granzyme B was observed shortly after treatment in CCC patients, followed by an increase in the number of cells expressing these molecules in a longer time after treatment. (p<0.05 for granzyme B).

Furthermore, the proportions of antigen-specific CD4+ T cells that express 1, 2, 3, 4 or 5 of these molecules under study (represented by the colors of the pie charts, Fig 6) were measured.

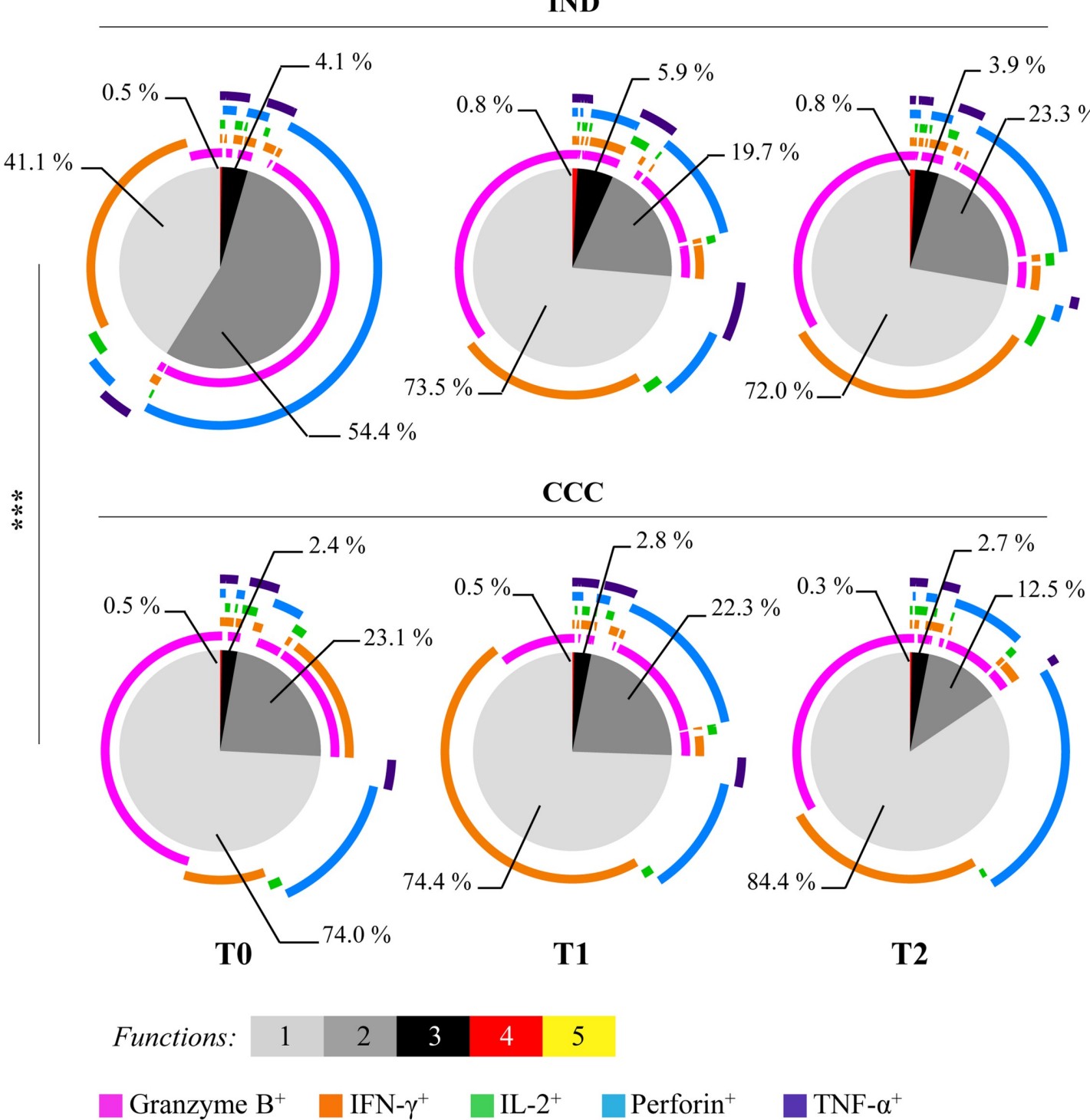

**Fig 6. Longitudinal evaluation of multifunctional antigen-specific CD4⁺ T cells from indeterminate and cardiac chronic Chagas disease patients evaluated before and after treatment.** Representation of the multifunctional profile (cytokines: IFN-γ, IL-2, and TNF-α and cytotoxic molecules: perforin and granzyme B) of CD4⁺ T cells generated in response to *T. cruzi* soluble antigens. The colors of the pie chart represent the proportion of cells expressing 1 or coexpressing 2, 3, 4 and 5 of the molecules under study. The pie chart arcs show the proportion of cells that express each evaluated molecule. PBMC of each patient were stimulated with *T. cruzi* soluble antigens (*Tc*SA) in presence of CD28 and CD49d co-stimulators for 10 h. The data shown were calculated by subtracting to these values those obtained with cells cultured only with CD28 and CD49d (background). Nineteen indeterminate patients (IND) and 13 Chagas disease patients with cardiac alterations (CCC) formed the study population at T0, and 17 out of 19 (IND) and 12 out of 13 patients (CCC) at T1 (9–12 months after treatment) and T2 (24–48 months after treatment) were analyzed. The analysis to compare pie charts was performed using SPICE software version 5.3 by generating 10,000 permutations. The p values are indicated by asterisks (p<0.05 (*) and p<0.001 (***)).

The results showed that after 9–12 months of treatment, IND showed an increase in the proportion of cells expressing 1 (from 41.1% at T0 to 73.5% at T1), 3 (from 4.1% at T0 to 5.9% at T1) and 4 (from 0.5% at T0 to 0.8% at T1) molecules (Fig 6, panel T0 and T1). However, a reduction in the proportion of cells that coproduced two molecules was observed after 9–12 months of treatment (from 54.4% to 19.7%) in IND patients. The multifunctional pattern observed in IND patients at a short time after treatment was maintained after 24–48 months of BNZ administration (Fig 6, panel T2). Furthermore, we evaluated which particular molecules were expressed or coexpressed at each tested time point (color arcs of the pie charts, Fig 6). Thus, the increase in the proportion of CD4+ T cells from IND patients that showed the expression of only 1 molecule was produced as a result of the rise in the secretion of granzyme B. Notably, the increase in the subset of CD4+ T cells that expressed 3 molecules after treatment was mainly reflected in cells that coproduced both the cytotoxic molecules under study and IFN-γ (CD4+granzyme B+perforin+ and IFN-γ+), which consequently showed a Th1 profile (1.4% before treatment and 4.5% at 9–12 months after treatment). On the other hand, the subset of CD4+ T cells from IND that coproduced granzyme B and perforin (CD4+granzyme B+perforin+IFN-γ−IL-2−TNF-α−) decreased after 9–12 months of treatment (from 49.8% to 10.4%) and showed a slight increase after 24–48 months (to 16.8%). Conversely, CCC showed no significant changes in terms of the multifunctional pattern of CD4+ T cells after 9–12 months of treatment (Fig 6, panel CCC, T0 *versus* T1 or T2). The proportion of CD4+ T cells from CCC patients that coproduced 3 molecules was increased after 9–12 months of treatment with BNZ (from 2.4% to 2.8%), and this increase was maintained at 24–48 months after treatment (2.7%). Moreover, the proportion of the CD4+ T cell subset that coexpressed granzyme B and IFN-γ (15.8%) decreased after treatment (2.5% at T1), and conversely, the proportion of the CD4+ T cell subset that coexpressed the two cytotoxic molecules under study (perforin and granzyme B) increased markedly after treatment (from 3.1% at T0 to 15.5% at T1). The proportion of antigen-specific CD4+ T cells expressing IFN-γ was increased at the follow-up performed before and after treatment in both the IND (28.4% at T0, 23.4% at T1, and 32.0% at T2) and CCC groups (10.0% at T0 to 48.1% at T1 and 25.0% at T2) (Fig 6, panel CCC).

## Effect of benznidazole on the parasite load and clinical signs

Presence of *T. cruzi* parasite in blood samples of the IND and CCC patients was analyzed by conventional PCR. Before treatment, the obtained results showed a positive PCR in 12 out of 22 IND (54.6%) and 8 out of 14 CCC (57.1%) patients (Table 1). From 9 to 12 months after treatment, 1 out of 19 IND patients (5.3%) had a positive PCR result. None of the 13 CCC patients analyzed had a positive PCR at 9–12 months after treatment. No parasites were detected in any of the IND and CCC patients analyzed at 24–48 months posttreatment since all had a negative PCR result (Table 1). Throughout the study, no clinical changes were observed in the patients.

## Discussion

Persistent exposure to pathogenic antigens during chronic infection with several intracellular parasites triggers a process of cellular dysfunction in antigen-specific T cells [14,34]. In the context of chronic *T. cruzi* infection, it has been reported that a high percentage of CD8+ T cells and the CD4+CD8+ T cell population show a gradual increase in the expression and coexpression of inhibitory receptors associated with a loss of function, which suggests that these T cell subsets undergo an exhaustion process [15,16,31]. In the present manuscript, it is shown that the CD4+CD8− T cell population undergoes a dysfunctional process during chronic infection caused by *T. cruzi*. A fraction of circulating CD4+ T cells from chronic Chagas disease

patients upregulate the expression of different inhibitory receptors (2B4, CD160, CTLA-4, PD-1 and TIM-3), as determined by comparison with the expression data obtained from healthy donors. Notably, the upregulation of PD-1 inhibitory receptor expression on CD4+ T cells from chronic patients with *T. cruzi* infection was the most marked increase in terms of frequency and expression level observed in comparison with healthy donors. PD-1 was the first discovered and most well-known exhaustion marker and has been widely shown to be involved in viral chronic infections [35–37], other disorders such as cancer [38,39] and parasite infections [40,41]. The expression of inhibitory receptors is considered a hallmark of exhaustion; these molecules often exhibit synergistic activity, and the exhaustion process becomes more dramatic when different inhibitory receptors and a larger number of molecules per cell are coexpressed [14,37,42]. CD4+ T cells from patients with chronic *T. cruzi* infection presented a markedly increased coexpression level of inhibitory receptors compared to CD4+ T cells from healthy donors. The T cell exhaustion process has been reported to occur gradually, and the highest coexpression of inhibitory receptors is accompanied by the greatest loss in functional capacities in a stage known as the severe exhaustion process [14,37,42,43]. Furthermore, our results show the highest coexpression level of inhibitory receptors in CD4+ T cells was observed in chronic Chagas disease patients with cardiac disorders, whose also presented a predominantly monofunctional antigen-specific T cell response. Thus, the CD4+ T cell population from the CCC patients showed a higher exhaustion status that those from indeterminate patients. A similar behavior was reported in *T. cruzi*-specific CD8+ T cells from chronic Chagas disease patients [15,16]. Therefore, we hypothesized that this dysfunctional process, which occurs in an important fraction of the T cell subset, could be involved in the loss of the control of *T. cruzi* infection and perhaps could be associated with the enhancement of the symptomatology. However, as has been reported in viral chronic infections [44,45], the CD4+ T cell dysfunction process detected in chronic Chagas disease patients presented divergences and similarities compared to the similar process observed in CD8+ T cells. Thus, the frequency of CD4+ T cells expressing the 2B4+ or CD160+ inhibitory receptors is significantly lower than that reported in the CD8+ T cell population from the same cohort of patients (4.4% and 1.5% in CD4+ T cells *versus* 40.0% and 10.0% in CD8+ T cells, respectively) [16]. Furthermore, the frequency of CD4+CTLA-4+ cells found in the present study was 24.3%, while in CD8+ T cells, the reported frequency of cells expressing CTLA-4 was significantly lower (approximately 7.0%) [15,16]. A similar situation has been described for the CD4+ and CD8+ T cell subsets from HIV patients [46,47]. Thus, the majority of exhausted CD8+ T cells from HIV patients showed expression of 2B4 and CD160, while the preferential expression of CTLA-4 was described for CD4+ T cells from these patients.

Chemotherapy with benznidazole causes a marked decline in the coexpression levels of inhibitory receptors in CD4+ T cells, which could indicate a partial reversion of the dysfunction process in this cell population. A similar effect of anti-parasitic treatment on CD8+ T cells from patients in the indeterminate form of Chagas disease [16,22] and on HIV-specific CD4+ T cells after ART treatment has also been reported [43]. Our findings show that the decline in inhibitory receptor coexpression after benznidazole treatment occurs more markedly in indeterminate Chagas disease patients than in Chagas patients with cardiac alterations. This could be linked to the observation of the highest inhibitory receptor coexpression level in cardiac patients. In line with this, Wherry et al. reported that the potential to reinvigorate the exhaustion process might be linked to the magnitude of exhaustion [48]. It has been reported that cells with PD-1hi expression were not rescued compared to those with PD-1mid expression [49], and other studies also demonstrated that blocking more than one inhibitory receptor pathway results in a synergistic reversal of the exhaustion process [42,50]. Furthermore, the highest impact of benznidazole treatment observed in our study was related to the expression

levels of the PD-1 marker by CD4+ T cells, which markedly declined after treatment in indeterminate and cardiac Chagas disease patients. These findings are crucial since the decrease in the expression of the PD-1 marker has been related to the improvement of disease, as reported in HIV-treated patients, in whom a decrease in PD-1 expression levels in HIV-specific CD4+ T cells was directly correlated with the plasma viral load [51]. In addition, blocking the PD-1/PD-1 L pathway augmented the functionality of LCMV-specific CD4+ T cells in a chronic model of infection, resulting in the increased production of IFN-γ, an increase in the frequency of multifunctional CD4+ T cells and the improvement and recovery of exhausted CD8+ T cells, which resulted in a striking reduction in the viral load [52]. Moreover, the expression of TIM-3 and CTLA-4 increased after treatment with benznidazole, regardless of the context of coexpression. This increase could be related to the dual function described for these markers, which act as both coinhibitory and costimulatory molecules [53,54]. Interestingly, when we evaluated the particular subsets of cells that were increased in indeterminate patients after treatment, we discovered increases in the percentages of CD4+ T cells that expressed only CTLA-4 and both CTLA-4+ and TIM-3+. These subsets of cells could correspond to regulatory T cells, since such cells have been shown to express the CTLA-4 receptor [55]; more recently, a population of regulatory T cells that express TIM-3 has been characterized as showing improved enhancer functions [56–58].

Regarding the results obtained after treatment, a statistically significant decrease in the coexpression of inhibitory receptors in the population of CD4+ T cells was observed, which occurred markedly in the short-to-medium term. Simultaneously, with this decline, a modulation of the functional antigen-specific CD4+ T cell response occurred (as measured by the production of molecules in response to parasite antigens), with an increase in the frequency of multifunctional antigen-specific CD4+ T cells and an increase in the proportion of CD4+IFN-γ+ T cells; this corresponds to a Th1 profile that is related to improved infection control [7,59]. These results could indicate that the reversal of the dysfunction process, as measured by the coexpression of inhibitory receptors, modulates the functional response of antigen-specific CD4+ T cells. The modifications observed in the immune response were maintained up to 24–48 months post treatment, the end time of the present study. The functional profile of the CD4+ T cells was significantly different between IND and CCC groups (p<0.001). Thus, a higher proportion of CD4+ T cells coexpressing 2 and 3 of the studied molecules was found in IND than in CCC while the percentage of monofunctional *T. cruzi*-specific CD4+ T cells was much higher in the CCC than in IND. Remarkably, after treatment these proportions acquired profiles more homogeneous between IND and CCC. However, after treatment the percentage of CD4+ T cells expressing each one of the molecules under study showed substantial differences in IND *versus* CCC. Thus, shortly after treatment the percentage of CD4+ T cells from IND patients showed an increase in the expression of IFN-γ and granzyme B while in CCC there was a decrease of the percentage of the CD4+ T cells expressing IFN-γ, granzyme B and IL-2. The results obtained regarding the expression of cytokines and cytotoxic molecules together the multifunctional capacity of the antigen-specific CD4+ T cells showed that treatment induced in chronic Chagas disease patients (IND and CCC) an early (9–12 months posttreatment) and substantial changes in the expression and coexpression of the molecules under study. The data obtained also supported that this new expression pattern was temporarily maintained for at least 24–48 months after treatment, the time of completion of the present study. Furthermore, the correlation studies carried out regarding the expression of each of the inhibitory receptors before treatment and in the short and long-term posttreatment indicated the existence of a better association between the T1 and T2 data (S1 Table), which means that the changes produced by the treatment are maintained over time. The functional immune response and its modulation after treatment described in this manuscript was observed in the

blood compartment. The variations than can take place in other compartments and tissues may be different. In fact it has been described that a Th1 response in the cardiac tissue is associated to a severe cardiac pathology. It has been also reported that heart lesions from CCC patients present a Th1 T-cell-rich myocarditis, with cardiomyocyte hypertrophy and prominent fibrosis [60].

Additionally, we hypothesized that perhaps an impact on the functionality of CD4+ T cells could be observed, and, perhaps in a more remarkable way, after a short time posttreatment. Studies of the production of IFN-γ and IL-2 by CD4+ T cells in chronic disease patients after treatment has allowed us to observe that some of the patients presented a peak in the production of these cytokines after a short time (2 to 6 months), which was followed by a drop or another peak in production after long-term treatment; in both cases, this was correlated with a drop in the antibody production rate due to the clearance of the parasite [19]. Perhaps a similar behavior occurred in our study cohort, as shown by the production of similar molecules (IFN-γ and IL-2) and other molecules (TNF-α, perforin and granzyme B). Previous studies carried out on circulating leukocytes before and one-year after benznidazole treatment after *in vitro* stimulation with live *T. cruzi* organisms support that the drug elicits a complex phenotypic/functional network compatible with beneficial and protective immunological events [61]. It has been also demonstrated that Bz-treatment led to higher activation status of circulating monocytes in addition to an elevated activation status associated with a type 1-modulated cytokine pattern [62]. These results suggest that Bz-treatment may also involve a qualitative change in the leukocytes functional capacity that drives their activation state toward a modulated cytokine profile [62].

On the other hand, it could be possible that the decrease in exhausted CD4+ T cells could contribute to an improvement in the functionality of CD8+ T cells after treatment through the helper function of CD4+ T cells [52,63,64]. Thus, the results in the present manuscript show the existence of a very high proportion of antigen-specific CD4+granzyme B+perforin+ T cells in IND *versus* CCC. The effector subpopulation of CD8+ T cells with cytotoxic capacities, known as CTLs, has been widely described in *T. cruzi* and other infections [65,66]. In contrast, a subset of CD4+ T cells with cytotoxic capability was identified as an unexpected CD4 CTL population. These CD4+ T cells can secrete cytotoxic granules containing granzyme B and perforin and directly kill target cells in an antigen-specific fashion, especially under chronic infection conditions [67,68]. In the case of chronic influenza virus infection, antigen-specific CD8+ CTL activity is impaired by exhaustion, and CD4+ CTLs act instead of these cells to contribute to infection control [69]. In the context of chronic Chagas disease, little is known about this subset of cells. Kesse et al. found increased expression of granzyme B and the degranulation factor CD107a in CD4+ T cells after *T. cruzi* stimulation in IND. These cytotoxic markers were also positively correlated with a memory profile [70]. The aforementioned reported findings, which are linked to our results, suggest that cytotoxic CD4+ T cells could play a key role in the control of chronic *T. cruzi* infection and in the outcome of the pathology, especially in patients who present an exhausted CD8+ T cell population. Remarkably, the present work showed the modulation of this CD4+ T cell subset after treatment with benznidazole, which caused an increase in the frequency of CD4+ T cells that coexpressed perforin and granzyme B in CCC patients and a reduction in the IND group. Furthermore, we found an increased proportion of CD4+granzyme-perforin+ T cells in CCC *versus* IND Chagas disease patients and that CCC had a CD4+granzyme B+INF-γ+ T cell population, which was absent in IND. This subset could be related to a reduced cytotoxic activity or may be implicated in tissue damage. In this sense, the single production of these cytotoxic molecules has been reported as less efficient in knock-out experimental models [71,72]. Moreover, the production of a single perforin could play a detrimental role in *T. cruzi*-elicited cardiac injury, as reported [73,74].

Analysis of circulating parasites indicated that prior to benznidazole treatment, 54.6% and 57.1% of IND and CCC patients had a positive PCR for *T. cruzi* detection. After 9–12 months of benznidazole treatment, only one IND patient (5.3%) had a positive PCR. No parasites were detected in any of the IND and CCC patients at 24 and 48 months after treatment. These results support a beneficial effect of benznidazole on the load of circulating parasites as it was previously reported in other studies [75,76]. We estimate that the persistence of the parasite load led to an exhaustion of the functional specific response of the T cells. Consequently, it is expected that a drop in the parasite load will recovery the multifunctional response partially associated to a drop of the coexpression level of the inhibitory receptors.

Treatment with benznidazole causes a decrease in the coexpression of inhibitory receptors that leads to an improvement in the multifunctional capacity of antigen-specific CD4+ T cells. However, the analysis of the individual expression of each inhibitory receptor showed that the expression of some of them decreased after treatment while the expression of others increased, as it was the case of CTL4 and TIMT-3. Therefore, this modulation seems to be aimed at maintaining the homeostasis of the immune response, which will result in an increase in the multifunctional activity of antigen-specific CD4+ T cells that will allow the elimination of the parasite avoiding an exacerbated immune response that could produce organ damage. In fact, therapies aimed at blocking a single inhibitory pathway may not achieve modification of the functional capacity of the cells or reach a perceptible improvement of any of the antigen-specific functions of partially exhausted cells [77]. However, few investigations that block these signaling pathways have been performed in cells from chronic Chagas disease patients. In this context, it has been described that CTLA-4 blocking does not produce a quantitative increase in antigen-specific IFN-γ production [78].

In summary, the data provided related to the study of the functional profiles of T cell populations could serve as a potential tool to predict the prognosis of chronic Chagas disease patients and be useful in monitoring the efficacy of therapy with current drugs in chronic patients and evaluating the future implementation of therapies, which is a great need for the proper care of this neglected tropical disease.

## Supporting information

**S1 Fig. Gating strategy used in flow cytometry analyses to evaluate the expression of the molecules under study by the CD4+ T cell population.** From left to right: **A)** Selection of singlets was performed on a dot plot that faces height *versus* area (FSC-H/FSC-A). Single lymphocytes were gated using FSC and SSC parameters. Subsequently, live cells were selected by its negative fluorescence after LIVE/DEAD Aqua dead cell staining. **(B)** Selection of CD3+ cells among the CD4+CD8-. **(C)** Gating strategy for selection of antigen-specific CD4+ T cells expressing IL-2, IFN-γ and TNF-α cytokines or granzyme B and perforine cytotoxic molecules. **(D)** Gates to assess the frequency of CD4+ T cells expressing inhibitory receptors (2B4, CD160, CTLA-4, PD-1 and TIM-3). The analyses were performed using the FlowJo 9.3.2 software. The selection of positivity for each marker was selected based on the fluorescence minus one (FMO) control, and the unstained control.
(TIF)

**S2 Fig. Evaluation of the production and coproduction of cytokines and cytotoxic molecules by CD4+ T cells from patients with chronic Chagas disease against *Trypanosoma cruzi* soluble antigens.** The frequency of *T. cruzi* antigen-specific CD4+ T cells that express or coexpress different numbers of molecules of IFN-γ, IL-2, TNF-α, perforin and/or granzyme B was evaluated in 19 indeterminate patients (IND) and 13 patients with cardiac manifestations (CCC). The coexpression analyses were performed using a Boolean gates strategy, of the five

molecules under study. The data shown were obtained after stimulation with *T. cruzi* soluble antigens (*Tc*SA) plus CD28 and CD49d for 10 h, subtracting the background value (cells cultured with CD28 and CD49d co-stimulators). The lines included in the graphs represent the mean for each group of data. Statistical analyzes were performed using the Mann-Whitney U test. The p values are indicated by asterisks (p<0.05 (*)).
(TIF)

**S3 Fig. Functional capacity of *T. cruzi*-specific CD4+ T cells from indeterminate and cardiac chronic Chagas disease patients, before and after treatment.** Frequency of *Tc*SA-estimulated CD4+ T cells producing the IFN-γ, IL-2, and TNF-α cytokines, or the perforin and granzyme B cytotoxic molecules, from19 indeterminate patients (IND) and 13 Chagas disease patients with cardiac alterations (CCC) before treatment (T0); from 17 out of 19 (IND) and 12 out of 13 (CCC) at 9–12 (T1) and 24–48 (T2) months after treatment. The data shown were obtained after stimulation of cells with *T. cruzi* soluble antigens (*Tc*SA) plus CD28 and CD49d for 10 h and subtracting to this value the background value (cells cultured solely with CD28 and CD49d). Statistical analyzes were performed using the Wilcoxon test. Statistically significant differences (p<0.05) are indicated with an asterisk (*). Scatter plots showing the median values represented with horizontal lines.
(TIF)

**S1 Table. Spearman's rank matrix showing a correlation study between the expression of the different inhibitory receptors under study and different time points [pretreatment (T0) and 9–12 (T1) and 24–48 (T2) months post benznidazole treatment].**
(PDF)

## Acknowledgments

We appreciate the participation of patients and healthy volunteers that allowed the realization of this study. This publication is part of the PhD thesis of Elena Pérez Antón at the University of Granada in the Biomedicine Program.

## Author Contributions

**Conceptualization:** Manuel Carlos López.

**Data curation:** Bartolomé Carrilero, Miguel Ángel López-Ruz.

**Formal analysis:** Elena Pérez-Antón, Adriana Egui, Manuel Carlos López.

**Funding acquisition:** M. Carmen Thomas, Manuel Carlos López.

**Investigation:** Elena Pérez-Antón, Adriana Egui, M. Carmen Thomas, Marina Simón, Manuel Carlos López.

**Methodology:** Elena Pérez-Antón, Adriana Egui, Marina Simón.

**Resources:** M. Carmen Thomas, Bartolomé Carrilero, Miguel Ángel López-Ruz, Manuel Segovia, Manuel Carlos López.

**Supervision:** Manuel Segovia, Manuel Carlos López.

**Validation:** M. Carmen Thomas, Manuel Carlos López.

**Visualization:** Elena Pérez-Antón, Adriana Egui, M. Carmen Thomas.

**Writing – original draft:** Elena Pérez-Antón, Manuel Carlos López.

**Writing – review & editing:** Elena Pérez-Antón, Adriana Egui, M. Carmen Thomas, Manuel Carlos López.

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
