## [Decision Letter · Decision Letter 0]

22 Sep 2020

Dear PhD Lopez,

Thank you very much for submitting your manuscript "A proportion of CD4+ T cells from patients with chronic Chagas disease undergo a dysfunctional process, which is partially reversed by benznidazole treatment." for consideration at PLOS Neglected Tropical Diseases. As with all papers reviewed by the journal, your manuscript was reviewed by members of the editorial board and by several independent reviewers. In light of the reviews (below this email), we would like to invite the resubmission of a significantly-revised version that takes into account the reviewers' comments. 

The manuscript was evaluated by three experts in Chagas disease which requested further description and clarification of many points in methods, results and discussion. In addition, we agree with the referee #1 that representative plots of gating strategy and main findings are crucial for a paper that relies on flow cytometry for its major conclusion. Therefore, we request the authors to clarify the points raised by referees and provide gating strategy and representative plots of flow cytometry for the main findings.

We cannot make any decision about publication until we have seen the revised manuscript and your response to the reviewers' comments. Your revised manuscript is also likely to be sent to reviewers for further evaluation.

Sincerely,

Helton da Costa Santiago, M.D., Ph.D

Associate Editor

Eric Dumonteil

Deputy Editor

The manuscript was evaluated by three experts in Chagas disease which requested further description and clarification of many points in methods, results and discussion. In addition, we agree with the referee #1 that representative plots of gating strategy and main findings are crucial for a paper that relies on flow cytometry for its major conclusion. Therefore, we request the authors to clarify the points raised by referees and provide gating strategy and representative plots of flow cytometry for the main findings.

Reviewer's Responses to Questions

**Key Review Criteria Required for Acceptance?**

**Methods**

-Are the objectives of the study clearly articulated with a clear testable hypothesis stated?

-Is the study design appropriate to address the stated objectives?

-Is the population clearly described and appropriate for the hypothesis being tested?

-Is the sample size sufficient to ensure adequate power to address the hypothesis being tested?

-Were correct statistical analysis used to support conclusions?

-Are there concerns about ethical or regulatory requirements being met?

Reviewer #1: The manuscript “A proportion of CD4+ T cells from patients with chronic Chagas disease undergo a dysfunctional process, which is partially reversed by benznidazole treatment” by Elena Pérez-Antón et al. approaches a relevant question and brings very interesting data concerning the dysfunction of CD4+ T cells (using inhibitory receptors markers and cytokines) and the therapeutic role of benznidazole (Bz), a quite important topic. However, some questions were raised and shall be clarified to facilitate the comprehension of the results. 

General comments:

The cohort and the parameters used to classify the patients shall be better described.

The effects of Bz on parasite load shall be shown and the correlation with the modulation of inhibitory molecules or reduction of the frequency of cells expression these receptors. 

The experimental protocols shall be better described.

The results shall be better shown, including representative profiles of FACS analysis. 

The discussion shall consider the temporal and compartmentalization of the immune response. 

The putative modulatory effects of Bz shall be better discussed

Reviewer #2: The objectives of the study are clearly articulated with a clear testable hypothesis stated, the study design is appropriate to address the stated objectives, the population is clearly described and appropriate for the hypothesis being tested, the sample size is sufficient to ensure adequate power to address the hypothesis being tested. My only concern about the methods is that T2 time of blood collection involves 24 to 48 months after treatment. This is such a long time variation. The authors should explain why is that.

Also, can you describe better your antigen? Is it excreted and secreted products of trypos?

Reviewer #3: The objectives and hypothesis is clearly presented and the study design is appropriate to address them.

The sample size is sufficient.

**Results**

-Does the analysis presented match the analysis plan?

-Are the results clearly and completely presented?

-Are the figures (Tables, Images) of sufficient quality for clarity?

Reviewer #1: Abstract – 

1- Asymptomatic and symptomatic are not synonymous od IND and CCC. A patient with the cardiac form may be asymptomatic. Therefore, this point shall be revised, and a unique nomenclature adopted all the manuscript. 

2- The analyzed molecules shall be described in the abstract. 

3- The conclusion shall be revised, or the data reanalyzed as correlation studies were not performed and there is no analysis of correlation with the severity of the CCC form (mild, moderate, severe). 

4- Was the beneficial effect of Bz transient? Is it?

5- A more general conclusion shall be proposed. 

Introduction

1- Line 77: Conceptually, is Chagas disease spread or infected persons? 

2- The description of the data is superficial. The literature findings supporting the selection of the biomarkers used shall be better sustained, the analyzed phenotypes described. 

3- As the present study is not the first to approach the effects of Bz therapy on Tcell functional biomarkers, therefore some data should be presented in this section. 

Material and Methods 

1- Patients are neither material nor methods. A section of Description of the cohort is required, including the ethics statements. 

2- Were all the studied samples submitted to the same protocol? Collected, cells separated, frozen? Were samples matched? For example, time kept frozen? 

3- How many simultaneous labelling were performed? What were the combination of staining? (ex CD3, CD4, x, y,z, w). What was the apparatus used? how many cells How were the samples analyzed and the software used? The gating strategies shall be described [for example, singlets (R1), dead-cell exclusion (FSC-A × SSC-Lin, R2), CD3 × CD4 dot plot, so on] 

4- Line 189: Why was this time used? The authors did not justify the choice of the time point (10 hours) used in cells cultures.

5- Do all the biomarkers show the same kinetics of expression/regulation/modulation? Is 10 hs the appropriate time to study all the biomarkers?

6- Line 192: The use of anti-CD28 is clear. But why CD49d was used? (the chose of these antibodies shall be support by literature)

7- Line 198: It is not clear: 10 hs of stimulus and 9 hs with brefeldin? Why this?

8- How was followed the effectiveness of Bz effects on parasite load? These data shall be presented. Any effects on clinical signs?

Results

1- In general, results shall be better described.

2- The sequence of cytometry analysis and the representative profiles of the FACS analysis shall be shown for all the studied profiles (singlets, R1, dot plots). 

3- Are all data generated of the analyses of results obtained after antigen stimulation (antigen + CD28/CD49d)? Were the results obtained after anti-CD28/CD49d stimulation considered to express the final data (graphs)? Are there differences of results according to these two stimulations?

4- The legends of the figures shall be improved, describing the stimulators, time of stimulus, way of calculation and statistical method used.

5- All the description using IND/asymptomatic and CCC/symptomatic shall be revised (abstract, introduction, results, legends, discussion). The authors should consider that absence (IND) or presence of signs of cardiac disease (CCC) after exams; the classification of symptomatic or asymptomatic is based on descriptions of the patient. A patient with the cardiac form of Chagas disease may auto declare to be asymptomatic. 

6- Figure 6: The pie charts are remarkably interesting and shall be maintained. However, the frequencies of cytokines and the profiles of cytotoxic markers are required and should be presented (as the previous figures), allowing the comparison with other findings (including the literature). Representative plots are also needed to support the findings. 

7- Is there a relation with the profiles and CCC severity (mild/ moderate, severe)- correlation graphs may help to clarify this question.

Reviewer #2: The analysis presented matched the analysis plan and the results are clearly and completely presented. The figures are of good quality, with exception of Fig 5.

Figure 5 has a word in Spanish (leyend) which should be translated to English. Also, I would love to see this figure bigger. The small size is not helping in appreciating the data. Maybe you could divide it in 5 graphs, according to the number of molecules expressed, or find another way of representing it.

Reviewer #3: I suggested some adjustments for the data presentation.

**Conclusions**

-Are the conclusions supported by the data presented?

-Are the limitations of analysis clearly described?

-Do the authors discuss how these data can be helpful to advance our understanding of the topic under study?

-Is public health relevance addressed?

Reviewer #1: Discussion

1- Line 486/487: To state this “correlation” (…a positive correlation between the degree of exhaustion and the severity of the pathology was found), the study shall be performed. 

2- After the analysis and correlation analyses, the discussion shall be rewritten. 

3- The discussion shall consider the temporal and compartmentalization of the immune response. 

4- The putative modulatory effects of Bz shall be better discussed and supported.

Conclusion and limitations should be added.

Reviewer #2: The conclusions are supported by the data presented and the public health relevance is well addressed.

However, the authors could discuss further their data using different points of view of the complex pathogenesis of this disease. Suggestions can be found below.

Reviewer #3: The conclusions is not completely supported by the data presented and the limitations of analysis are not clearly described. I suggest some adjustments in the text.

**Editorial and Data Presentation Modifications?**

Reviewer #1: Figures shall be revised, data included.

Reviewer #2: Minor revision.

Please include the discussion suggested and consider changing figure 5 for a better appreciation. 

Also, if possible, PCR data from blood of these patients showing parasite load could considerably enrich your data.

Reviewer #3: No comments.

**Summary and General Comments**

Reviewer #1: The manuscript “A proportion of CD4+ T cells from patients with chronic Chagas disease undergo a dysfunctional process, which is partially reversed by benznidazole treatment” by Elena Pérez-Antón et al. approaches a relevant question and brings very interesting data concerning the dysfunction of CD4+ T cells (using inhibitory receptors markers and cytokines) and the therapeutic role of benznidazole (Bz), a quite important topic. However, some questions were raised and shall be clarified to facilitate the comprehension of the results. 

General comments:

The cohort and the parameters used to classify the patients shall be better described.

The effects of Bz on parasite load shall be shown and the correlation with the modulation of inhibitory molecules or reduction of the frequency of cells expression these receptors. 

The experimental protocols shall be better described.

The results shall be better shown, including representative profiles of FACS analysis. 

The discussion shall consider the temporal and compartmentalization of the immune response. 

The putative modulatory effects of Bz shall be better discussed.

Reviewer #2: The manuscript is well written, clearly and carefully presented. The data are great. It touches the very exciting topic about why some people develop symptomatic forms of the disease, while others remain asymptomatic. That's the great one-billion dollar question about Chagas disease. 

I have concerns about the author summary, though. In line 54 the authors state that "Trypanosoma cruzi infection triggers several immune mechanisms in the host that are mostly inadequate to achieve the total clearance of the parasite". This is clearly the view of the authors. This reviewer thinks you should change it to a more scientific, impersonal description of what happens: "the immune mechanisms in the host that don't result in a total clearance of the parasite", or something like that. The same is suggested for the word "superior" (line 60). It could be changed for something more accurate or just suppressed. 

A question arises and could enrich your discussion.

Is the decline in the co-expression of inhibitory receptors really advantageous for the host? Isn't a regulated response ideal? What if the cells in the Benznidazole-treated indeterminate patients are now more capable of responding and, in an attempt to kill the remaining parasites, also damage the organs and development of symptomatic chronic disease is now triggered? 

Are these patients been followed? Will they progress slower than untreated indeterminate patients or not progress to disease at all? 

I miss data showing parasite load on these patients on times T0, T1 and T2. Maybe a PCR in the blood. I know the blood is not so reliable to show parasite load, but it's worth trying.

The authors must be familiar that it's been suggested that the host organism's goes for maintaining a chronic disease for homeostatic purposes. Too low response results in death with high parasitism. Too big response results in death by immunopathology. So, the body goes for a middle term. Balance. Of course there is a price to be paid in order to keep that population alive for the next generations, and this price happens to be 25% of the individuals dying at the end of the road with cardiac failure. Inhibitory receptors and other regulatory mechanisms are also a means to achieve a balance between low level immunity and low level parasitism. There is this fear that intervening with these regulatory mechanisms in asymptomatic, presumably balanced, patients, could somehow move the scale towards enhanced inflammation and, therefore, disease. 

Which brings me to another suggestion. Please discuss the mechanism whereby Benznidazole treatment causes the decline on the co-expression of the inhibitory receptors. Is it because of reduced parasitism? Can you show the reduced parasitism by PCR? 

On the other hand, if proved that the decline in the expression of inhibitory receptors is actually beneficial, so each and every asymptomatic patient should be treated as soon as the infection is diagnosed.

So, in this manuscript, you beautifully show the association between clinical form and expression of these inhibitory receptors. It remains to be shown the real contribution of these "more capable" cells for a symptom-free form of the infection. Would they be capable of totally clearing the parasite?

I think all of those points should be addressed in your discussion in a concise way. You've done that with regard of perforin and granzyme positive CD8 T cells and you can extend the discussion to the inhibitory receptors. 

Another interesting point is the one that arises on line 111-114: "Signs of senescence, the later stages of differentiation, the effects of immunoregulatory mechanisms and the expression of inhibitory receptors associated with more severe forms of Chagas disease have also been described in the CD4+ T cell population". How is that in the symptomatic individuals we have worse immunopathology even with all that? Is this the result of increased, uncontrolled parasitism? Can it be shown? It can be shown in the animal model, perhaps. Of course this reviewer don't think you need to show this, but it's an interesting point to be discussed.

Reviewer #3: General comments: The manuscript presented by Pérez-Antón and colleagues is mainly focused on the evaluation of CD4+ T cells from patients with chronic Chagas disease in asymptomatic or cardiac/symptomatic phase to determine the expression/coexpression of inhibitory molecules and how the antigen-specific multifunctional capacity of these cells is affected. In addition, the authors also evaluated the potential impact of benznidazole (Bz) treatment on the Trypanosoma cruzi-specific functional response of CD4+ T cells and their coinhibitory molecule´s expressions.

The main findings demonstrated an increase in the multifunctional capacity of the antigen specific CD4 + T cells in asymptomatic in comparison with cardiac patients, which was associated with the reduced coexpression of inhibitory receptors. Moreover, after short-term Bz treatment, asymptomatic patients showed a significant increase in the frequency of multifunctional antigen specific CD4 + T cells.

The manuscript is relevant and provides new insights about how functional profiles of T cell populations may serve as a potential tool to predict the prognosis of chronic Chagas disease patients and be useful in monitoring the efficacy of therapy in chronic Chagas patients. However, I have some concerns as listed below.

Specific comments:

- In the Abstract – Conclusions/Significance – I suggest excluding the statement that “cell dysfunctional process… was correlated with the severity of the pathology”, since this correlation analysis was not performed in the manuscript.

- I suggest throughout the text substitute the term “indeterminate” for “asymptomatic” to standardize the clinical definition of the chronic Chagas disease.

- I suggest substituting the title of the topic “Material and methods” by “Population, material and methods”.

- In the topic Study subjects, the authors should be inserted the demographic data (age ranging and gender) from patients and whether they present any co-morbidities (eg. hypertension, diabetes, coagulopathies, etc…).

- Why the authors opt to use the Y strain of T. cruzi to obtain the parasite soluble antigens?

- The concentration of TcSa (1 μg/mL) employed in the short-term in vitro antigen stimulation of PBMC seems very low. Please, clarify this choice.

- For the flow cytometry assays is not clear what was the cytometer used to acquire the samples, the software used to analyze the data, and the number of events acquired/sample. Please, clarify these issues in the Material and Methods.

- Regarding the gating strategy used to analyze the data, I suggest including a complementary figure to present in details this strategy using representative density plots. 

- The number of patients mentioned in the Results usually is not the same described in the Study subjects topic. Please, clarify.

- In general, for the results, there are a large variability among the values in many parameters evaluated. Are the data responsible for this variability provided by the same patients? Please, clarify this issue in the text.

- Considering the data presented in the Figure 3, did the authors performed any correlations between expression of inhibitory receptors and distinct timepoints after Bz treatment? This analysis may reinforce the findings obtained iusing a categorical analysis.

- The statement about the Figure 5 presented in the text, lines 367 to 370 is confuse. Please, clarify the issue considering the magnitude of increase/decrease in the frequency of inhibitory receptors in asymptomatic patients.

- In the Figure 6, before Bz treatment, the data demonstrated that the proportion of cells expressing 1 or coexpressing 2, 3 is predominant and there are distinct profiles in these proportions in the comparison between asymptomatic and CCC groups. After treatment, these proportions acquired profiles more homogeneous between both groups. These findings must be more broadly addressed in the Discussion.

- In some circumstances, the text of Discussion is very speculative, and the authors did not show sufficient data to fundament the statements: e.g. Lines 479-480 - “These findings could indicate that a fraction of the CD4+ T cell population undergoes a T cell exhaustion process…”; Lines 486-487 - “Thus, a positive correlation between the degree of exhaustion and the severity of the pathology was found”. Please, revise them.

- The limitations of the study must be stated in the Discussion.

- In the lines 406-407, the acronym for interferon-gamma (IFN-y) is incorrectly presented in the text as “INF-γ”. Please, revise it.

PLOS authors have the option to publish the peer review history of their article (what does this mean?). If published, this will include your full peer review and any attached files.

Reviewer #1: No

Reviewer #2: No

Reviewer #3: No
---

## [Decision Letter · Decision Letter 1]

22 Dec 2020

Dear PhD Lopez,

We are pleased to inform you that your manuscript 'A proportion of CD4+ T cells from patients with chronic Chagas disease undergo a dysfunctional process, which is partially reversed by benznidazole treatment.' has been provisionally accepted for publication in PLOS Neglected Tropical Diseases.

Best regards,

Helton da Costa Santiago, M.D., Ph.D

Associate Editor

Eric Dumonteil

Deputy Editor

Reviewer's Responses to Questions

**Key Review Criteria Required for Acceptance?**

**Methods**

-Are the objectives of the study clearly articulated with a clear testable hypothesis stated?

-Is the study design appropriate to address the stated objectives?

-Is the population clearly described and appropriate for the hypothesis being tested?

-Is the sample size sufficient to ensure adequate power to address the hypothesis being tested?

-Were correct statistical analysis used to support conclusions?

-Are there concerns about ethical or regulatory requirements being met?

Reviewer #1: The revised version is improved and answered the main points and requests.

Reviewer #2: (No Response)

Reviewer #3: All requirements above mentioned were accomplished.

**Results**

-Does the analysis presented match the analysis plan?

-Are the results clearly and completely presented?

-Are the figures (Tables, Images) of sufficient quality for clarity?

Reviewer #1: Figures and Table were added. Results were rewritten.

Reviewer #2: (No Response)

Reviewer #3: After review, the results are clearly presented and the figures present sufficient quality for clarity.

**Conclusions**

-Are the conclusions supported by the data presented?

-Are the limitations of analysis clearly described?

-Do the authors discuss how these data can be helpful to advance our understanding of the topic under study?

-Is public health relevance addressed?

Reviewer #1: The text was revised following suggestions.

Reviewer #2: (No Response)

Reviewer #3: All requirements above mentioned were accomplished.

**Editorial and Data Presentation Modifications?**

Reviewer #1: none

Reviewer #2: (No Response)

Reviewer #3: No comments.

**Summary and General Comments**

Reviewer #1: The revised version is improved and answered the main points and requests. Figures and Table were added. Results were rewritten. Conclusions were revised following suggestions. The manuscript was improved and may be considered for publication.

Reviewer #2: In my opinion, the quality of the manuscript has considerably improved after the revision.

I just want to point out 2 small typing errors: Line 754 - "Benznidazole" instead of Benznidaole; Line 759 - "recover", instead of recovery.

Reviewer #3: The authors considered the Reviewers requests and made the alterations required or suggested. The quality of manuscript has been satisfactorily improved in the revised version of the manuscript.

PLOS authors have the option to publish the peer review history of their article (what does this mean?). If published, this will include your full peer review and any attached files.

Reviewer #1: **Yes: **Joseli Lannes-Vieira

Reviewer #2: No

Reviewer #3: No

---

## [Editor Report · Acceptance letter]

29 Jan 2021

Dear PhD Lopez,

We are delighted to inform you that your manuscript, "A proportion of CD4+ T cells from patients with chronic Chagas disease undergo a dysfunctional process, which is partially reversed by benznidazole treatment.," has been formally accepted for publication in PLOS Neglected Tropical Diseases.

Best regards,

Shaden Kamhawi

co-Editor-in-Chief

Paul Brindley

co-Editor-in-Chief
